# Investigation of a *Legionella pneumophila* Outbreak at a Bath Facility in Japan Using Whole-Genome Sequencing of Isolates from Clinical and Environmental Samples

**DOI:** 10.3390/microorganisms11010028

**Published:** 2022-12-22

**Authors:** Noriko Nakanishi, Shoko Komatsu, Shinobu Tanaka, Kengo Mukai, Ryohei Nomoto

**Affiliations:** Department of Infectious Diseases, Kobe Institute of Health, 4-6-5 Minatojima-nakamichi, Chuo-ku, Kobe 650-0046, Japan

**Keywords:** *Legionella pneumophila*, outbreak, bath facility, Japan, Legionnaires’ disease, whole-genome sequencing

## Abstract

Exposure to aerosols containing *Legionella* from artificially made water systems has been established as a primary cause of Legionnaires’ disease. In this study, we investigated an outbreak of *L. pneumophila* serogroup 1 sequence type 138 which occurred at a bath facility in 2022. The whole-genome sequencing of isolates revealed that the colonization of *L. pneumophila* at the bath facility had occurred before 2013, and the patients had been exposed to multiple genetic lineages of the strain. Our study demonstrates the importance of performing a careful comparative genetic analysis of clinical and environmental isolates from LD outbreaks in order to effectively investigate and prevent future LD outbreaks.

Legionnaires’ disease (LD) outbreaks occur when people are exposed to water contaminated with *Legionella* [1]. *Legionella pneumophila* serogroup 1 (SG1) accounts for most cases of *Legionella* infections in humans [2]. In Japan, hot springs and public bath facilities are major sources of *Legionella* infection [3]. In this study, we report the first case identified as an outbreak of LD caused by *L. pneumophila* SG1 sequence type (ST) 138 based on whole-genome sequencing, which occurred at a bath facility in Japan.

In March 2022, two cases of LD were reported among users of a bath facility in Kobe City, Hyogo Prefecture. The facility consisted of two bath facilities each (one of which was a bubble bath facility) for men and women. Both the bubble bath facilities were connected to a common circulation system which included filtration and heating components.

Sputum samples from two patients (who tested positive for urinary antigens detecting *L. pneumophila*), one on day 4 and the other on day 8 of illness onset, and environmental samples from the bath water, hair-catcher and the filtration components in the bath facility were collected for epidemiological investigations and tested for the presence of *Legionella* at our laboratory. *Legionella* could not be detected in the sputum sample of one patient in PCR and culture tests because the sputum was obtained after 8 days of illness onset and the administration of antibiotics; however, 14 *L. pneumophila* strains were isolated from the sputum sample of the other patient, which was cultured in a selective medium (MWY medium; Kanto Chemical Co., Inc., Tokyo, Japan). The serogrouping of *Legionella* isolates was performed using slide agglutination tests with monovalent antisera (Denka Seiken, Tokyo, Japan). Sequence-based typing (SBT) was performed according to the protocol of the European Working Group for Legionella infections (EWGLI) [4,5]. All the strains isolated from this patient sample belonged to SG1 ST138. The samples collected from the bubble bath water, hair catcher and filtration components also contained *L. pneumophila* SG1 ST138.

*Legionella pneumophila* SG1 ST138, detected only in Japan, belongs to the B3 group (bathwater group) and leads to sporadic cases and small outbreaks of LD [6]. To investigate the genetic similarities between different strains identified in this outbreak, we performed whole-genome sequencing to compare the 14 strains of *L. pneumophila* isolated from the patient sputum sample and those isolated from the environmental samples collected from the bath facility (5 strains each isolated from the bath water and hair-catcher samples and 8 strains isolated from the sample collected from the filtration components). All the isolates were found to harbor *lag-1*, a virulence-associated marker [7]. Whole-genome sequencing data were obtained using the MiSeq instrument (Illumina). Single-nucleotide variant (SNV) analysis was performed using BactSNP [8], and the removal of the recombinant region was achieved using Gubbins [9]. The median mapping depth of the analyzed samples to the reference genome (*L. pneumophila* str. Paris, Accession no.; CR628336.1) was 167.

The strains isolated from the patient sputum sample and environmental samples from the outbreak facility differed by 0–42 SNVs. Haplotype network analysis based on SNVs showed that 12 of the 14 strains isolated from the patient sputum sample could be divided into clades I, II and III, each differing from the other by 10 or fewer SNVs (Figure 1). Clade I consisted of only five patient strains with 0–1 SNVs. Clades II and III contained both strains from the patient sample and environmental samples with 3-4 and 2 SNVs, respectively (clade II (four strains from the patient sample and two strains from the bath water sample); clade III (three strains from the patient sample and five strains from swabs of the hair catcher)). Isolates from other LD outbreaks have been shown to differ from each other by as few as five core SNVs [10]. Therefore, the results of this study confirmed that the bath facility was indeed the source of the *Legionella* infection in this outbreak. Environmental strains that were nearly identical to the two patient strains, KL2256 and KL2240, could not be isolated through our environmental survey. However, these genetic subtypes were supposed to be potentially present in the bath facility. Thus, it can be concluded that the patient was exposed to multiple genetic subtypes of *Legionella* belonging to ST138, as reported earlier [11,12].

Whole-genome sequencing of *L. pneumophila* SG1 ST138 strain KL0954 isolated from the sample collected from the bubble bath facility of the same bath house in 2013 was also conducted. The difference between the strains collected from the patient sample and KL0954 was approximately 30 SNVs. Furthermore, KL0954 differed from the KL2300 and KL2301 strains isolated from the filtration components’ sample by only nine SNVs and differed from the KL2220 strain isolated from the bath water sample by ten SNVs. Epidemiological and laboratory investigations revealed that the inadequate maintenance of the filtration components, bubble generators (which had not been replaced and cleaned for more than 10 years) and circulation systems resulted in *Legionella* colonization in the bath facilities. Taken together, it is possible that *L. pneumophila* SG1 ST138 strains had colonized prior to 2013 and had diversified into multiple genetic lineages within the bath facility.

Our investigation into this outbreak showed that the bath facility was colonized by different genetic subtypes of *L. pneumophila* SG1 ST138. Therefore, genetic comparisons between multiple clinical isolates (from one patient) and environmental isolates (from the site of an outbreak) should be carried out to effectively investigate and prevent future LD outbreaks and determine whether patients have been exposed to multiple genetic strains of *Legionella*.

## Figures and Tables

**Figure 1 microorganisms-11-00028-f001:**
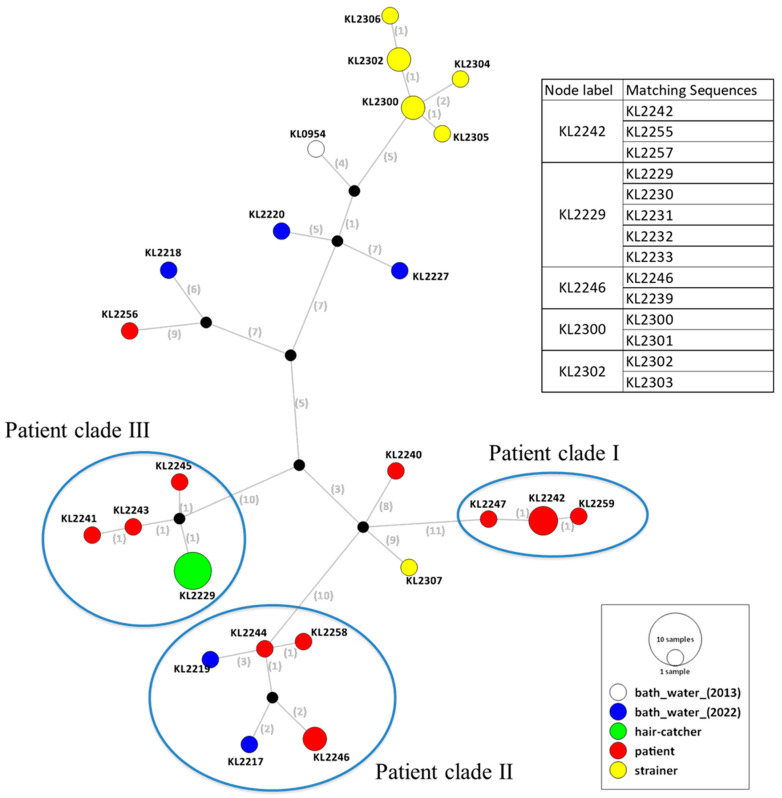
Haplotype network analysis based on single-nucleotide variants (SNVs) using whole-genome sequencing of 33 *Legionella pneumophila* serogroup 1 (SG1) sequence type (ST) 138 strains. The numbers between each node represent the number of SNVs. The strains isolated from the patient and environmental samples, bath water, hair catcher and filtration are shown as red, blue, green and yellow circles, respectively. KL0954 isolated from bath water in 2013 is indicated as white circle. The raw sequence data used in this study were deposited in DDBJ/EMBL/GenBank under DRA accession number DRR410368-DRR410400.

## Data Availability

All new WGS data of *L. pneumophila* strains obtained as part of this study are deposited in the DDBJ/EMBL/GenBank under BioProject accession number PRJDB14217. The DRA accession numbers are DRR410368-DRR410400.

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
