# Peer review of "Investigation of a Legionella pneumophila Outbreak at a Bath Facility in Japan Using Whole-Genome Sequencing of Isolates from Clinical and Environmental Samples"

_microorganisms, 2022, doi:10.3390/microorganisms11010028_

Round 1
Reviewer 1 Report
The authors present herein a case of an outbreak investigation where they tried to link human cases with environmental source using whole genome sequencing. The authors conclude that the patients had been exposed to multiple genetic lineages of Legionella and they demostrate the importance of performing comparative genetic analysis of clinical and environmental isolates to effectively investigate and detect the source of infection.
The manuscript is generally well writen however, there are several points into the text that the authors need to expand, such as the methodology part and the results presentation. Certainly more info is required to help the authors understand the whole process from sampling to source identification.
I have attached a pdf file with some of my comments that the authors may find helpful.

Author Response
Response to Reviewer 1:
We have changed in revised manuscript according to your advice (Paragraph 3).
Comments: The authors claim above that the bath facility was the source of infection. How can this be the case if the environmental strains and the human isolates do not match?
Response: Thank you for this comment. As you indicated, we were unable to isolate strains closely related to the two patient strains (KL2256 and KL2240) through our environmental survey. However, since we have not been able to isolate all of the genetic lines present in this bath facility, we believe it is probable that these genetic subtypes of patient strains were also potentially present in this bath facility. On the other hand, since the patient and environmental strains in Clades II and III differ by only a few SNVs, we can conclude that this facility is the source of infection.
Comments: The authors should explain how they ended up at this conclusion
Response: Thank you for this comment. Three strains, KL2300, KL2301 and KL2220, were isolated that differed by only 10 SNVs or less from the strain isolated in 2013. Considering the inadequate maintenance of the facility, where the filtration components and bubble generators had not been replaced and cleaned for more than 10 years, it is possible that the isolates had been colonized in the facility since 2013. In accordance with Reviewer’s comment, we have changed it to a milder expression, referring to the possibilities found in these results as follows text.
“Whole-genome sequencing of L. pneumophila SG1 ST138 strain KL0954 isolated from the sample collected from the bubble bath facility of the same bath house in 2013 was also conducted. The difference between strains collected from the patient sample and KL0954 was approximately 30 SNVs. Furthermore, KL0954 differed from the KL2300 and KL2301 strains isolated from the filtration components sample by only nine SNVs, and differed from the KL2220 strain isolated from the bath water sample by 10 SNVs. Epidemiological and laboratory investigation revealed that inadequate maintenance of filtration components, bubble generators (not replaced and cleaned for more than 10 years), and circulation systems resulted in Legionella colonization in the bath facilities. Taken together, it is possible that L. pneumophila SG1 ST138 strains had colonized prior to 2013, and had diversified into multiple genetic lineages within the bath facility.”
Reviewer 2 Report
The manuscript form Nakanishi et al. present the genetic analysis of 33 strains of L. pneumophila isolated in the context of a case of Legionnaires' disease linked to a bath facility. The authors shows that the isolates differed by a few SNVs and suggest the presence of diverse variants in the environment an in the sputum.
Overall the manuscript is well written and easy to follow. I have only minor comments that would improved th reproducibility of the manuscript and strengthen the conclusions.
My main concern is that the authors do not present any quality control data about the assembled genomes. This is an issue as the SNV analysis is very sensitive to low quality assembly. I suggest to clarify how the raw sequence reads were processed and assembled and provide quality control data, such as N50, number of contigs, dead-ends, etc. This could be summarize in the manuscript and included as supplementary material.
Paragraph 3: how was serogroup and sequence typing done. Was the later done by sequencing of PCR amplicon (ESGLI protocol) or was it derived form the WGS?
Paragraph 6: "The strains isolated from the environmental samples..." is confusing, as Figure 1 shows that none of the patient strains are identical to the environmental strains.
Paragraph 7: Since the author only sequenced 1 strains from 2013, the diversity of ST138 at that moment is currently unknown. It is therefore difficult to conclude that since then "...ST138 had diversified into multiple genetic lineages within the bath facility". I agree that this is likely what happened, but the data do not support this very well.
I believe that this study also shows that when using WGS to match patient and environmental strains, one should probably tests more than one strains from each, as to be able to estimate diversity within the system. Maybe the author should comment on this.
Abstract: "... containing Legionella in artificially" change "in" to "from".
Paragraph 1: extra "." at the end of paragraph.
Figure 1: since there was only 1 patient, remove "s" in "Patients Clade"
Author Response
Response to Reviewer 2:
Comments: My main concern is that the authors do not present any quality control data about the assembled genomes. This is an issue as the SNV analysis is very sensitive to low quality assembly. I suggest to clarify how the raw sequence reads were processed and assembled and provide quality control data, such as N50, number of contigs, dead-ends, etc. This could be summarize in the manuscript and included as supplementary material.Response: In the analysis described in this paper, genome sequence assembly was not performed, but SNV analysis was performed based on mapping data against a reference sequence. Therefore, there are no N50 or contig count data that should be included in this paper. However, when the assembly was performed separately from the paper, the average N50 was approximately 78240 bp and the median contig count was 127, so we believe that there are no problems with data quality.
In accordance with Reviewer’s comment, we have added a description of the mapping in the text (page 2, 1st paragraph) as follows.
“The median mapping depth of the analyzed samples to the reference genome (L. pneumophila str. Paris, Accession no.; CR628336.1) was 167”
Comments: Paragraph 3: how was serogroup and sequence typing done. Was the later done by sequencing of PCR amplicon (ESGLI protocol) or was it derived form the WGS?
Response: We understand the concern of Reviewer in this regard. In accordance with Reviewer’s comment, we have added a description of the method of serogroup and sequence typing in the text as follows. References have been renumbered since references were added for SBT.
“Serogrouping of Legionella isolates was performed using a slide agglutination tests with monovalent antisera (Denka Seiken, Tokyo, Japan). Sequence-based typing (SBT) was performed according to the protocol of the European Working Group for Legionella infections (EWGLI) [4, 5].”
Comments: Paragraph 6: "The strains isolated from the environmental samples..." is confusing, as Figure 1 shows that none of the patient strains are identical to the environmental strains.
Response: We thank the reviewer for pointing this out. We have combined paragraphs 5 and 6 into a single paragraph, and changed the following text.
“It could not be isolated environmental strains that were nearly identical to the two patient strains, KL2256 and KL2240 through our environmental survey. However, these genetic subtypes was supposed to be potentially present in the bath facility. Thus, it can be concluded that the patient was exposed to multiple genetic subtypes of Legionella belonging to ST138, as reported earlier [11, 12].”
Comments: Paragraph 7: Since the author only sequenced 1 strains from 2013, the diversity of ST138 at that moment is currently unknown. It is therefore difficult to conclude that since then "...ST138 had diversified into multiple genetic lineages within the bath facility". I agree that this is likely what happened, but the data do not support this very well.
Response: Thank you for this comment. Three strains, KL2300, KL2301 and KL2220, were isolated that differed by only 10 SNVs or less from the strain isolated in 2013. We also found the diversity of strains isolated from one patient and the inadequate maintenance at the facility at least since 2013. However, as the reviewer pointed out, since only one isolate from 2013 was add to analyze, we could not investigate the process of divergence. Therefore, we have changed it to a milder expression in paragraphs 7, referring to the possibilities found in these results as follows text.
‘Whole-genome sequencing of L. pneumophila SG1 ST138 strain KL0954 isolated from the sample collected from the bubble bath facility of the same bath house in 2013 was also conducted. The difference between strains collected from the patient sample and KL0954 was approximately 30 SNVs. Furthermore, KL0954 differed from the KL2300 and KL2301 strains isolated from the filtration components sample by only nine SNVs, and differed from the KL2220 strain isolated from the bath water sample by 10 SNVs. Epidemiological and laboratory investigation revealed that inadequate maintenance of filtration components, bubble generators (not replaced and cleaned for more than 10 years), and circulation systems resulted in Legionella colonization in the bath facilities. Taken together, it is possible that L. pneumophila SG1 ST138 strains had colonized prior to 2013, and had diversified into multiple genetic lineages within the bath facility.”
Comments: I believe that this study also shows that when using WGS to match patient and environmental strains, one should probably tests more than one strains from each, as to be able to estimate diversity within the system. Maybe the author should comment on this.
Response: Our conclusion is the same idea as you have indicated. In particular, we believe that it is necessary to analyze multiple strains derived from the one patient, but since this was not clear in our conclusion, we have revised it as follows highlighted text.
“Therefore, genetic comparison between multiple clinical isolates (from one patient) and environmental isolates (from the site of an outbreak) should be carried out to effectively investigate and prevent future LD outbreaks and determine whether patients were exposed to multiple genetic strains of Legionella.”
Comments: Abstract: "... containing Legionella in artificially" change "in" to "from".
Response: We have changed in the revised manuscript according to your advice as follows.
Comments: Paragraph 1: extra "." at the end of paragraph.
Response: We appreciate the reviewer’s pointing out. We revised as suggested if the revised manuscript.
Comments: Figure 1: since there was only 1 patient, remove "s" in "Patients Clade"
Response: We appreciate the reviewer’s pointing out. We revised as suggested if the revised manuscript.